# Comparative Analysis of Nonlinear Programming Solvers: Performance Evaluation, Benchmarking, and Multi-UAV Optimal Path Planning

**Giovanni Lavezzi** [1,*] **, Kidus Guye** [2] **, Venanzio Cichella** [3] **and Marco Ciarcià** [4]

1 Department of Aeronautics and Astronautics, Massachusetts Institute of Technology, Cambridge, MA 02139, USA
2 Department of Mechanical Engineering, University of Maryland, College Park, MD 20742, USA; kguye@umd.edu
3 Department of Mechanical Engineering, University of Iowa, Iowa City, IA 52242, USA; venanzio-cichella@uiowa.edu
4 Department of Mechanical Engineering, South Dakota State University, Brookings, SD 57006, USA; marco.ciarcia@sdstate.edu
* Correspondence: glavezzi@mit.edu

**Abstract:** In this paper, we propose a set of guidelines to select a solver for the solution of nonlinear programming problems. We conduct a comparative analysis of the convergence performances of commonly used solvers for both unconstrained and constrained nonlinear programming problems. The comparison metrics involve accuracy, convergence rate, and computational time. MATLAB is chosen as the implementation platform due to its widespread adoption in academia and industry. Our study includes solvers which are either freely available or require a license, or are extensively documented in the literature. Moreover, we differentiate solvers if they allow the selection of different optimal search methods. We assess the performance of 24 algorithms on a set of 60 benchmark problems. We also evaluate the capability of each solver to tackle two large-scale UAV optimal path planning scenarios, specifically the 3D minimum time problem for UAV landing and the 3D minimum time problem for UAV formation flying. To enrich our analysis, we discuss the effects of each solver's inner settings on accuracy, convergence rate, and computational time.

**Keywords:** NLP; unconstrained; constrained; UAV; path planning; optimization





## 1. Introduction

The current technological era prioritizes, more than ever, high performance and efficiency of complex processes controlled by a set of variables. Examples of these processes are [1–8]: engineering designs, chemical plant reactions, manufacturing processes, grid power management, power generation/conversion process, path planning for autonomous vehicles, climate simulations, etc. Quite often, the search for the best performance, or the highest efficiency, can be transcribed into a nonlinear programming (NLP) problem, namely, the need to minimize (or maximize) a scalar cost function subjected to a set of constraints. In some instances, these functions are linear but in general, one or both of them are characterized by nonlinearities. For simple one-time use problems, one might successfully use any of the solvers available, such as FMINCON in MATLAB [9,10]. Nevertheless, if the NLP derives from some specific applications, such as real-time process optimization, then the solver choice begs a more accurate selection.

The first research efforts toward the characterization of optimization solvers began in the 1960s. In [11], the authors compare eight solvers on twenty benchmark unconstrained NLP problems containing up to 20 variables. Notably, they illustrate techniques to transform particular constrained NLPs into equivalent unconstrained problems. The authors of [12] analyze the convergence properties of two gradient-based solvers applied

to 16 test problems. In the last few decades, with the development of new methodologies and optimization applications, more studies aimed to illustrate difference in performance among NLP solvers. Schittkowski et al. [13] performed the comparison of eleven different mathematical programming codes applied to structural optimization through finite element analysis. George et al. summarize a qualitative comparison of few optimization methodologies reported by several other sources [14]. In the research document prepared by Sandia National Laboratory [15], a study was conducted on four open source Linear Programming (LP) solvers applied to 201 benchmark problems. In [16], Kronqvist et al. carried out a performance comparison of mixed integer NLP solvers limited to convex benchmark problems. The work in [17] presents a comparison between linear and nonlinear programming techniques on the diet formulation for animals. In [18], the authors use the programming language R to analyze multiple nonlinear optimization methods applied to real-life problems. State-of-the-art optimization methods were used to compare their application on L1-regularized classifiers [19]. On a similar note, multiple global optimization solvers were compared in a work done by Arnold Neumaier [20]. Authors from [21–24] conducted a performance comparison of optimization techniques for specific applications such as: aerodynamic shape design, integrated manufacturing planning and scheduling, solving electromagnetic problems, and building energy design problems, respectively. Similarly, Frank et al. conducted a comparison between three optimization methods for solving aerodynamic design problems [25]. In [26], Karaboga et al. compared the performance of the artificial bee colony algorithm with the differential evolution, evolutionary and particle swarm optimization algorithms using multi-dimensional numerical problems.

In this paper, we aim to provide an explicit comparison of a set of NLP solvers. In our comparison, we incorporate widely used solvers available in MATLAB, several gradient descent methods that have been extensively utilized in the literature, and a particle swarm optimization algorithm. Because of its widespread use among research groups, both in academia and the private sector, we have decided to use MATLAB as a common implementation platform. For this reason, we will focus on all the solvers that are either written on or can be implemented in MATLAB. The NLP problems used in this comparison have been selected amongst the standard benchmark problems [27–29] with up to thirty variables and up to nine scalar constraints.

In addition to the selected benchmark problems, two large-scale minimum-time unmanned aerial vehicle (UAV) optimal path planning problems are included in the analysis. UAVs represent a low-cost and effective alternative in many practical applications, such as precision agriculture, environmental monitoring, product deliveries, or military missions, and interest has increasingly growing in the past two decades [30]. UAV path planning is an important research area since it aims to improve their autonomous and safety capabilities. Path planning often employs optimal control formulations as a key strategy. Utilizing this method, one can generate trajectories that not only minimize a specified cost function, but also adhere to both vehicle-related and problem-specific constraints. When path planning/optimal control problems are "simple", they can be solved analytically using one of two classical methods: the Bellmann [31,32] and the Pontryagin methods [33]. However, when dealing with complex (e.g., large-scale, non-convex, nonlinear) optimal control problems, it is very hard or even impossible to find solutions analytically using one of these methods, and numerical methods must be sought. Numerical methods are based on discretizing/transcribing the optimal control problem into a finite-dimensional NLP problem, which can be solved using ready-to-use NLP solvers (e.g., FMINCON, SNOPT, SciPy, IPOPT). In this paper, we employ Bernstein polynomial approximations to solve UAV path planning problems by generating 3D trajectories, thereby converting optimal control problems into NLP problems [34]. The approach is inspired by prior research on optimal control using Bernstein polynomials [35,36], which demonstrated the efficient generation of feasible and collision-free trajectories for both single and multiple vehicles. Moreover, this approach can be applied to real-time safety-critical applications in complex environments.

Therefore, the analysis is further improved by evaluating the performance of the solvers in a realistic case scenario.

The paper is organized as follows: Section 2 describes the statement of unconstrained and constrained NLP problems; in Section 3, we briefly enumerate the NLP solvers included in our analysis, provide an overview of the different convergence metrics, and finally, carry out the solvers' implementations; the results of the comparison with the benchmark equations are discussed in Section 4; in Section 5, each solver is tested to solve two real-world UAV path planning optimal control problems. Finally, the main contributions of the paper are outlined in Section 6.

## 2. Nonlinear Programming Problem Statements

In general, a constrained NLP problem aims to minimize a nonlinear real scalar objective function with respect to a set of variables while satisfying a set of nonlinear constraints. If the problem entails the minimization of a function without the presence of constraints, then the problem is defined as unconstrained [37]. In the following section, the general forms of nonlinear unconstrained and constrained optimization problems are stated.

### 2.1. Unconstrained Optimization Problem
#### 2.1.1. Statement

Let $x \in \mathbb{R}^n$ be a real vector with $n \geq 1$ components and let $f : \mathbb{R}^n \to \mathbb{R}$ be a smooth function. Then, the unconstrained optimization problem is defined as

$$\min_{x \in \mathbb{R}^n} f(x). \tag{1}$$

#### 2.1.2. Optimality Conditions

Given a function $f(x)$ defined and differentiable over an interval $(a, b)$, the necessary condition for a point $x^* \in (a, b)$ to be a local maximum or minimum is that $f'(x^*) = 0$. This is also known as Fermat's theorem. The multidimensional extension of this condition states that the gradient must be zero at local optimum point, namely,

$$\nabla f(x^*) = 0. \tag{2}$$

Equation (2) is referred to as a first-order optimality condition.

### 2.2. Constrained Optimization Problem
#### 2.2.1. Statement

The constrained optimization problem is formulated as

$$\min_{x \in \mathbb{R}^n} f(x) \tag{3}$$

subject to

$$g_i(x) \leq 0, \quad i = 1, 2, \ldots, w, \tag{4}$$

$$h_j(x) = 0, \quad j = 1, 2, \ldots, l, \tag{5}$$

with $g_i(x)$ and $h_j(x)$ smooth real-valued functions on a subset of $\mathbb{R}^n$. Notably, $g_i(x)$ and $h_j(x)$ represent the sets of inequality constraints and equality constraints, respectively. The feasible set is identified as the set of points $x$ that satisfy just the constraints (Equations (4) and (5)). It must be pointed out that some of the solvers considered in this study are only able to support equality constraints. In these instances, we will introduce a set of slack variables $s_i$, and convert Equation (5) into the following set of equality constraints

$$g_i(x) + s_i^2 = 0, \quad i = 1, 2, \ldots, w. \tag{6}$$

Such necessary expedients will obviously induce more computational burden on the particular solvers affected by this constraint-type limitation, since the slack variables $s_i$ become additional optimization variables. In this scenario, the constrained optimization problem can be reformulated as

$$\min_{\boldsymbol{x},\boldsymbol{s}\in\mathbb{R}^n} f(\boldsymbol{x},\boldsymbol{s}) \tag{7}$$

subject to

$$g_i(\boldsymbol{x}) + s_i^2 = 0, \quad i = 1,2,\ldots,w, \tag{8}$$

$$h_j(\boldsymbol{x}) = 0, \quad j = 1,2,\ldots,l. \tag{9}$$

### 2.2.2. Optimality Conditions

The Karush–Kuhn–Tucker (KKT) conditions measure the first-order optimality for constrained problems [38]. These necessary conditions are defined as follows. Let the objective function $f$ and the constraint functions $g_i$ and $h_j$ be continuously differentiable functions at $\boldsymbol{x}^* \in \mathbb{R}^n$. If $\boldsymbol{x}^*$ is a local optimum and the optimization problem satisfies some regularity conditions [37], then there exist the two constants $\mu_i$ $(i = 1,\ldots,w)$ and $\lambda_j$ $(j = 1,\ldots,\ell)$, called KKT multipliers, such that the following four groups of conditions hold:

- Stationarity:

$$f(\boldsymbol{x}) : \nabla f(\boldsymbol{x}^*) + \sum_{i=1}^{m} \mu_i \nabla g_i(\boldsymbol{x}^*) + \sum_{j=1}^{\ell} \lambda_j \nabla h_j(\boldsymbol{x}^*) = \boldsymbol{0}. \tag{10}$$

- Primal feasibility:

$$g_i(\boldsymbol{x}^*) \leq 0, \text{ for } i = 1,\ldots,w. \tag{11}$$

$$h_j(\boldsymbol{x}^*) = 0, \text{ for } j = 1,\ldots,\ell. \tag{12}$$

- Dual feasibility:

$$\mu_i \geq 0, \text{ for } i = 1,\ldots,w. \tag{13}$$

- Complementary slackness:

$$\sum_{i=1}^{m} \mu_i g_i(\boldsymbol{x}^*) = 0. \tag{14}$$

## 3. NLP Solvers and Convergence Metrics

This section briefly introduces the 24 solvers included in our study, narrates the key implementation steps for each solver, and provides the description of the convergence metrics used in our analysis.

### 3.1. NLP Solvers Selection

The selection of the NLP solvers considered in this work is based on the following aspects. First of all, we only examine algorithms that are available in MATLAB. Secondly, we have included solvers that are either free source or, for commercial software, have a trial version. The following is a list of the 24 solvers and algorithms that we have included in the benchmark analysis:

- Accelerated Particle Swarm Optimization (APSO): an algorithm developed by Yang at Cambridge University in 2007, and based on swarm-intelligent search of the optimum [39]. Due to the nature of the algorithm, only constrained nonlinear programming problems can be solved.
- Branch and Reduced Optimization Navigator (BARON): a commercial global optimization software that solves both NLP and mixed-integer nonlinear programs (MINLP) by using deterministic global optimization algorithms of the branch and

bound search type [40–42]. It comes with embedded linear programming (LP) and NLP solvers, such as CLP/CBC [43], IPOPT [44,45], FilterSD [46] and FilterSQP [47]. BARON selects the NLP solver by default and may switch between other NLP solvers during the search based on problem characteristics and solver performance. To refer to the default option, the name BARON (auto) is chosen. For this study, we have acquired the monthly license of the software to be used in conjunction with the MATLAB interface. Such choice reflects the fact that the free demo version is characterized by some limitations, namely, it can only handle problems with up to ten variables, ten constraints, and it does not support trigonometric functions.

- FMINCON: a MATLAB optimization toolbox used to solve constrained NLP problems [37]. FMINCON provides the user the option to select amongst five different algorithms to solve nonlinear problems: Active-set , Interior-point, Sequential Quadratic Programming (SQP), Sequential Quadratic Programming legacy (SQP-legacy), and Trust region reflective. Four out of the five algorithms are implemented in our analysis as one of them, the Trust Region Reflective algorithm, does not support most of the constraints considered in our benchmark cases.

- FMINUNC: a MATLAB optimization toolbox used to solve unconstrained NLP problems [48]. In this case, FMINUNC gives the user the option of choosing between two different algorithms to solve nonlinear minimization problems [49]: Quasi-Newton and Trust region.

- Globally Convergent Method of Moving Asymptotes (GCMMA): is a modified version of the MMA that evaluates the global optimum value [50–52].

- Nonlinear Interior point Trust Region Optimization (ARTELYS KNITRO): a commercially available nonlinear optimization software package developed by Zienna Optimization since 2001 [53] for finding local solutions to both continuous and discrete optimization problems with integer or binary variables, with or without constraints [53,54]. In this work, the software free trial license is used, in conjunction with the MATLAB interface. Several algorithms are included in the software, such as Interior-point/Direct, Interior-point/CG, Active-set, and Sequential Quadratic Programming. Interior-point/CG mainly differs from the Interior-point/Direct algorithm because of the primal-dual KKT system solved using a projected conjugate gradient iteration [54].

- Mixed Integer Distributed Ant Colony Optimization (MIDACO): a global optimization solver that combines an extended evolutionary probabilistic technique, called the Ant Colony Optimization algorithm, with the Oracle Penalty method for constrained handling [55,56]. In this work, we have obtained a license; otherwise, it must be noted that the free trial version has a limitation, namely, that it does not support more than four variables per problem.

- Method of Moving Asymptotes (MMA): it solves nonlinear problem function by generating an approximate subproblem [51,52,57].

- Modified Quasilinearization Algorithm (MQA): the modified version of the Standard Quasilinearization Algorithm (SQA) [58,59]. The goal is the progressive reduction of the performance index. Convergence to the desired solution is achieved when the performance index $\tilde{Q} \leq \varepsilon_1$ or $\tilde{R} \leq \varepsilon_2$, with $\varepsilon_1$ and $\varepsilon_2$ small preselected positive constants, for the unconstrained and constrained case, respectively [60,61]. Regarding NLP problems, it must be noted that the MQA can only handle equality constraints. As a result, slack variables are introduced to convert the inequality constraints into equality constraints.

- PENLAB: a MATLAB free open source software package suitable for nonlinear, linear semidefinite, and nonlinear semidefinite optimization, based on a generalized Augmented Lagrangian method [62–65].

- Sequential Gradient-Restoration Algorithm (SGRA): a first-order NLP solver, characterized by a restoration phase, followed by a gradient phase [66,67]. The goal is the progressive reduction of the performance index. The performance index is expressed

by $\tilde{R}$, which includes both the feasibility index $\tilde{P}$, and the optimality index $\tilde{Q}$. Convergence is achieved when the constraint error and the optimality condition error are $\tilde{P} \leq \varepsilon_1, \tilde{Q} \leq \varepsilon_2$, respectively, with $\varepsilon_1, \varepsilon_2$ small preselected positive constants. It must be noted that only equality constraints can be handled by the SGRA. As a result, slack variables are introduced to convert the inequality constraints into equality constraints.

- Sparse Nonlinear OPTimizer (SNOPT): a commercial software package for solving large-scale optimization problems, linear and nonlinear programs [68]. In this paper, we use the free trial version of the software in conjunction with the MATLAB interface, which can be retrieved at [69].

- SOLNP: originally implemented in MATLAB to solve general nonlinear programming problems, characterized by nonlinear smooth functions in the objective and constraints [70,71]. Inequality constraints are converted into equality constraints by means of slack variables.

- Standard Quasilinearization Algorithm (SQA): the standard version of the QA, and it uses QA techniques for solving nonlinear problems by generating a sequence of linear problems solutions [58,59]. As the MQA, SQA can only handle equality constraints. As a result, slack variables are introduced to convert the inequality constraints into equality constraints.

Unless stated otherwise, the solvers and algorithms are freely available. Mathematical details, description and documentation, the most direct source to each solver and algorithm, and all the benchmark test functions can be found in [72]. For each of the test functions, dimension, domain and search space, objective function, constraints, and minimum solution are listed.

### 3.2. Convergence Metrics

The main goal of this study is to characterize the convergence performance, in terms of accuracy and computational time, of the different solvers under analysis. We have selected a number of benchmark NLPs and compared the numerical solutions returned by each solver with the true analytical solution. Moreover, considering that the choice of the initial guesses critically affects the convergence process, we want to assess also the capability to converge to the true optimum, rather than converging to a local minima or not converging at all. With this in mind, we define as converging robustness the quality of a solver to achieve the solution when the search process is initiated from a broad set of initial guesses randomly chosen within the search domain. Finally, to have an accurate assessment of the computational time, we require the solver to repeat the same search several times and average out the total CPU time. As a result, given $N$ benchmark test functions, $M$ solvers/algorithms, $K$ randomly generated initial guesses, and $Z$ repeated identical search runs, a total of $N \times M \times K \times Z$ runs must be executed.

The following performance metrics are in order:

- Mean error [%]:

$$\bar{E}_m = \frac{1}{N} \sum_{n=1}^{N} \bar{E}_n, \quad \bar{E}_n = \frac{1}{K} \sum_{k=1}^{K} E_k, \quad E_k = 100 \frac{|f(\boldsymbol{x}) - f(\boldsymbol{x}^*)|}{\max(|f(\boldsymbol{x}^*)|, 0.001)} \tag{15}$$

with $f(\boldsymbol{x})$ the benchmark test function evaluated at the numerical solution $\boldsymbol{x}$ provided by the solver, $f(\boldsymbol{x}^*)$ the benchmark test function evaluated at the optimal solution $f(\boldsymbol{x}^*)$, $E_k$ the error associated to the run from the $k$-th randomly generated initial guess, $\bar{E}_n$ the mean error associated to the $n$-th benchmark test function, and $\bar{E}_m$ the mean error delivered by the $m$-th solver. The biunivocal choice of the denominator of $E_k$ is based on the fact that some benchmark test functions at the optimal solution have zero value; in this case, a value of 0.001 is chosen instead as reference value.

- Mean variance [%]:

$$\bar{\sigma}_m = \frac{1}{N} \sum_{n=1}^{N} \sigma_n, \quad \sigma_n = \frac{1}{K-1} \sum_{k=1}^{K} |E_k - \bar{E}_n|^2 \tag{16}$$

where $\sigma_n$ is the variance correspondent to the $n$-th benchmark test function, and $\bar{\sigma}_m$ the mean variance delivered by the $m$-th solver.

- Mean convergence rate [%]:

$$\bar{\gamma}_m = \frac{1}{N} \sum_{n=1}^{N} \gamma_n, \quad \gamma_n = 100 \frac{K - K_{conv}}{K} \tag{17}$$

with $K_{conv}$ the number of runs (from a pool of $K$ distinct initial guesses) which successfully reach convergence for the $n$-th function, $\gamma_n$ the convergence rate for the $n$-th function, and $\bar{\gamma}_m$ the mean convergence rate delivered by the $m$-th solver. The convergence rate is computed considering successful a run that satisfies the converging threshold conditions $E_k \leq E_{max} = 5\%$, and $CPU_k \leq CPU_{max} = 10$ s, with $CPU_k$ the CPU time required to the run starting from the $k$-th initial guess.

- Mean CPU time [s]:

$$\overline{CPU}_m = \frac{1}{N} \sum_{n=1}^{N} \overline{CPU}_n, \tag{18}$$

$$\overline{CPU}_n = \frac{1}{Z} \sum_{z=1}^{Z} \overline{CPU}_z, \quad \overline{CPU}_z = \frac{1}{K} \sum_{k=1}^{K} CPU_k \tag{19}$$

where $\overline{CPU}_z$ is the mean CPU time per $z$-th repetition, $\overline{CPU}_n$ is the mean CPU time related to the $n$-th benchmark test function, and $\overline{CPU}_m$ is the mean CPU time delivered by the $m$-th solver.

### 3.3. Solvers Implementation

In this paper, we analyze the convergence performances of the different solvers in terms of robustness, accuracy, and CPU time. Considering that the user might decide to tune the convergence parameters to favor one of these metrics, we have decided to perform the comparison for three separate implementation scenarios: plug and play (P&P), high accuracy (HA), and quick solution (QS). The plug and play settings, as the name suggests, are the "out-of-the-box" settings of each solver. The high accuracy settings are based on more stringent tolerances and/or on a higher number of maximum iterations with respect to the plug and play settings. This tuning aims to achieve a more precise solution. Finally, the quick solution settings are characterized by more relaxed convergence tolerances, and a lower number of maximum iterations with respect to the plug and play settings. In this scenario, the algorithms should reach a less accurate solution but in a shorter time. In general, the objective function, its gradient, the initial conditions, the constraint function (for constrained problems only), and the solver options are elements which are inputted to each solver. The objective function gradient is not necessary for APSO, BARON, MIDACO, and SOLNP, but it is optional for FMINCON/FMINUNC and KNITRO. For GCMMA/MMA, SGRA, and SNOPT, the gradient of both the objective and constraint functions is necessary. MQA/SQA and PENLAB, in addition to these inputs, require the Hessian of the objective function. In the following subsection, details on each solver and on the three different solver settings per each solver are described. It must be noted that, in most cases, the settings' names here reported are the same as the solver's options names used in the code implementation. In this way, the reader can have a better understanding of which solver's parameter has been tuned.

### 3.3.1. APSO

The three settings considered in the analysis are reported in Table 1, where *no. particles* is the number of particles, *no. iterations* is the total number of iterations, and $\gamma$ is a control parameter that multiplies $\alpha$, one of the two learning parameters or acceleration constants, $\alpha$ and $\beta$, the random amplitude of roaming particles and the speed of convergence, respectively. APSO does also require the number of problem variables, *no. vars*, to be defined but this parameter is, obviously, invariant for the three settings.

**Table 1.** APSO settings.

| Settings | P&P | HA | QS |
|---|---|---|---|
| *no. particles* | 15 | 50 | 10 |
| *no. iterations* | 300 | 500 | 100 |
| $\gamma$ | 0.9 | 0.95 | 0.95 |

### 3.3.2. BARON

The three settings considered in the analysis are reported in Table 2, with $EpsA$ the absolute termination tolerance, $EpsR$ the relative termination tolerance, and $AbsConFeasTol$ the absolute constraint feasibility tolerance. Due to the limitations of the solver, trigonometric functions are not supported; for this reason, the following test functions are excluded in the analysis: A.2, A.3, A.4, A.5, A.17, A.18, A.26 for unconstrained problems, and B.2, B.5, B.8, for constrained problems (refer to Appendix in [72]).

**Table 2.** BARON settings.

| Settings | P&P | HA | QS |
|---|---|---|---|
| $EpsA$ | $10^{-6}$ | $10^{-10}$ | $10^{-3}$ |
| $EpsR$ | $10^{-4}$ | $10^{-10}$ | $10^{-3}$ |
| $AbsConFeasTol$ | $10^{-5}$ | $10^{-10}$ | $10^{-3}$ |

### 3.3.3. FMINCON/FMINUNC

The three settings considered in the analysis are reported in Table 3, with $StepTolerance$ the lower bound on the size of a step, $ConstraintTolerance$ the upper bound on the magnitude of any constraint functions, $FunctionTolerance$ the lower bound on the change in the value of the objective function during a step, and $OptimalityTolerance$ the tolerance for the first-order optimality measure.

**Table 3.** FMINCON/FMINUNC settings.

| Settings | P&P | HA | QS |
|---|---|---|---|
| FMINCON | | | |
| $StepTolerance$ | $10^{-10}$ | $10^{-10}$ | $10^{-6}$ |
| $ConstraintTolerance$ | $10^{-6}$ | $10^{-10}$ | $10^{-3}$ |
| $FunctionTolerance$ | $10^{-6}$ | $10^{-10}$ | $10^{-3}$ |
| $OptimalityTolerance$ | $10^{-6}$ | $10^{-10}$ | $10^{-3}$ |
| FMINUNC (quasi-newton) | | | |
| $StepTolerance$ | $10^{-6}$ | $10^{-12}$ | $10^{-6}$ |
| $FunctionTolerance$ | $10^{-6}$ | $10^{-12}$ | $10^{-3}$ |
| $OptimalityTolerance$ | $10^{-6}$ | $10^{-12}$ | $10^{-3}$ |
| FMINUNC (trust-region) | | | |
| $StepTolerance$ | $10^{-6}$ | $10^{-12}$ | $10^{-6}$ |
| $FunctionTolerance$ | $10^{-6}$ | $10^{-12}$ | $10^{-3}$ |
| $OptimalityTolerance$ | $10^{-6}$ | $10^{-6}$ | $10^{-3}$ |

### 3.3.4. GCMMA/MMA

The three settings considered in the analysis are reported in Table 4, where *epsimin* is a prescribed small positive tolerance that terminates the algorithm, whereas *maxoutit* is the maximum number of iterations for MMA, and the maximum number of outer iterations for GCMMA.

**Table 4.** GCMMA/MMA settings.

| Settings | P&P | HA | QS |
|---|---|---|---|
| *epsimin* | $10^{-7}$ | $10^{-10}$ | $10^{-3}$ |
| *maxoutit* | 80 | 150 | 30 |

### 3.3.5. KNITRO

The three settings considered in the analysis are reported in Table 5, where *xtol* and *ftol* are tolerances that terminate the optimization process if the relative change of the solution point estimate or of the objective function are less than that values, *opttol* and *opttol_abs* specify the final relative and absolute stopping tolerance for the KKT (optimality) error, and *feastol* and *feastol_abs* specify the final relative and absolute stopping tolerance for the feasibility error.

**Table 5.** KNITRO settings.

| Settings | P&P | HA | QS |
|---|---|---|---|
| *xtol* | $10^{-6}$ | $10^{-10}$ | $10^{-3}$ |
| *ftol* | $10^{-6}$ | $10^{-10}$ | $10^{-3}$ |
| *opttol* | $10^{-6}$ | $10^{-10}$ | $10^{-3}$ |
| *opttol_abs* | $10^{-6}$ | $10^{-10}$ | $10^{-3}$ |
| *feastol* | $10^{-6}$ | $10^{-10}$ | $10^{-3}$ |
| *feastol_abs* | $10^{-6}$ | $10^{-10}$ | $10^{-3}$ |

### 3.3.6. MIDACO

The three settings considered in the analysis are reported in Table 6, where *maxeval* is the maximum number of function evaluation. It is a distinctive feature of MIDACO that allows the solver to stop exactly after that number of function evaluation. It must be noted that another tunable parameter is *accuracy*, namely, the accuracy tolerance for the constraint violation, that is not considered in the settings since no beneficial effect has been found compared to *maxeval*.

**Table 6.** MIDACO settings.

| Settings | P&P | HA | QS |
|---|---|---|---|
| *maxeval* | 50,000 | 150,000 | 10,000 |

### 3.3.7. MQA

The three settings considered in the analysis are reported in Table 7, with $\varepsilon_1$ and $\varepsilon_2$ the prescribed small positive tolerances that allow the solver to stop, when the inequality $\tilde{Q} \leq \varepsilon_1$ or $\tilde{R} \leq \varepsilon_2$ is met. As mentioned in Section 3, regarding NLP problems, MQA can only handle equality constraints. As a result, slack variables are introduced to convert the inequality constraints into equality constraints. In this study, for all the three settings considered in the analysis, a value of 1 is chosen as initial guess for all the slack variables.

**Table 7.** MQA settings.

| Settings | P&P | HA | QS |
|---|---|---|---|
| $\varepsilon_1$ | $10^{-5}$ | $10^{-8}$ | $10^{-2}$ |
| $\varepsilon_2$ | $10^{-4}$ | $10^{-5}$ | $10^{-3}$ |

### 3.3.8. PENLAB

The three settings considered in the analysis are reported in Table 8, where *max_inner _iter* is the maximum number of inner iterations, *max_outer_iter* is the maximum number of outer iterations, *mpenalty_min* is the lower bound for penalty parameters, *inner_stop_limit* is the termination tolerance for the inner iterations, *outer_stop_limit* is the termination tolerance for the outer iterations, *kkt_stop_limit* is the termination tolerance KKT optimality conditions, and *unc_dir_stop_limit* is the stopping tolerance for the unconstrained minimization.

**Table 8.** PENLAB settings.

| Settings | P&P | HA | QS |
|---|---|---|---|
| *max_inner_iter* | 100 | 1000 | 25 |
| *max_outer_iter* | 100 | 1000 | 25 |
| *mpenalty_min* | $10^{-6}$ | $10^{-9}$ | $10^{-3}$ |
| *inner_stop_limit* | $10^{-2}$ | $10^{-9}$ | $10^{-1}$ |
| *outer_stop_limit* | $10^{-6}$ | $10^{-9}$ | $10^{-3}$ |
| *kkt_stop_limit* | $10^{-4}$ | $10^{-6}$ | $10^{-2}$ |
| *unc_dir_stop_limit* | $10^{-2}$ | $10^{-9}$ | $10^{-1}$ |

### 3.3.9. SGRA

The three settings considered in the analysis are reported in Table 9, with $\varepsilon_1$ the tolerance related to the constraint error $\tilde{P}$, and $\varepsilon_2$ the tolerance related to the optimality condition error $\tilde{Q}$. Considering that the SGRA can only treat equality constraints, slack variables are introduced to convert the inequality constraints into equality constraints. In this study, for all the three settings considered in the analysis, a value of 1 is chosen for all the slack variables.

**Table 9.** SGRA settings.

| Settings | P&P | HA | QS |
|---|---|---|---|
| $\varepsilon_1$ | $10^{-9}$ | $10^{-10}$ | $10^{-8}$ |
| $\varepsilon_2$ | $10^{-4}$ | $10^{-6}$ | $10^{-2}$ |

### 3.3.10. SNOPT

The three settings considered in the analysis are reported in Table 10, where *major _iterations_limit* is the limit on the number of major iterations in the SQP method, *minor _iterations_limit* is the limit on minor iterations in the QP subproblems, *major_feasibility _tolerance* is the tolerance for feasibility of the nonlinear constraints, *major_optimality _tolerance* is the tolerance for the dual variables, and *minor_feasibility_tolerance* is the tolerance for the variables and their bounds.

**Table 10.** SNOPT settings.

| Settings | P&P | HA | QS |
|---|---|---|---|
| *major_iterations_limit* | 1000 | 10,000 | 100 |
| *minor_iterations_limit* | 500 | 5000 | 100 |
| *major_feasibility_tolerance* | $10^{-6}$ | $10^{-12}$ | $10^{-3}$ |
| *major_optimality_tolerance* | $10^{-6}$ | $10^{-12}$ | $10^{-3}$ |
| *minor_feasibility_tolerance* | $10^{-6}$ | $10^{-12}$ | $10^{-3}$ |

### 3.3.11. SOLNP

The three settings considered in the analysis are reported in Table 11, with $\rho$ the penalty parameter in the augmented Lagrangian objective function, *maj* the maximum number of major iterations, *min* the maximum number of minor iterations, $\delta$ the perturbation parameter for numerical gradient calculation, and $\epsilon$ the relative tolerance on optimality and feasibility. During the HA scenario implementation, we learned that different convergence settings are required for unconstrained and constrained problems. This peculiarity might be induced by the stringent tolerances adopted in this scenario.

**Table 11.** SOLNP settings. Tuning values for the HA scenario are divided for unconstrained (left-side) and constrained (right-side) problems.

| Settings | P&P | HA | QS |
|---|---|---|---|
| $\rho$ | 1 | 1 | 1 |
| *maj* | 10 | 500\|10 | 10 |
| *min* | 10 | 500\|10 | 10 |
| $\delta$ | $10^{-5}$ | $10^{-10} \mid 10^{-6}$ | $10^{-3}$ |
| $\epsilon$ | $10^{-4}$ | $10^{-12} \mid 10^{-7}$ | $10^{-3}$ |

### 3.3.12. SQA

The three settings considered in the analysis are reported in Table 12, with $\varepsilon_1$ and $\varepsilon_2$ the prescribed small positive tolerances that allow the solver to stop, when the inequality $\tilde{Q} \leq \varepsilon_1$ or $\tilde{R} \leq \varepsilon_2$ is met. As mentioned earlier, SQA can only treat equality constraints. To overcome this limitation, slack variables are introduced to convert the inequality constraints into equality constraints. In this study, for all the three settings considered in the analysis, a value of 1 is chosen for all the slack variables.

**Table 12.** SQA settings.

| Settings | P&P | HA | QS |
|---|---|---|---|
| $\varepsilon_1$ | $10^{-5}$ | $10^{-8}$ | $10^{-2}$ |
| $\varepsilon_2$ | $10^{-4}$ | $10^{-5}$ | $10^{-3}$ |

## 4. Benchmark Test Functions Analysis

We present a collection of unconstrained and constrained optimization test problems that are used to validate the performance of the various optimization algorithms presented above for the different implementation scenarios. The comparison results are also discussed in depth in this section.

For performance comparison purposes, an equivalent environment and control parameters have been created to run each NLP solver. All outputs tabulated in this paper are calculated using MATLAB software running on a desktop computer with the following specs: Intel(R) Core(TM) i7-6700 CPU 3.40GHz processor, 16.0 GB of RAM, running a 64-bit Windows 10 operating system. To assess the true computational time required by each algorithm to reach convergence, implementation steps that are expected to have an impact on the computer's performance are deactivated during the run for the solution. The internet connection and other unrelated applications are turned off throughout the

analysis, ensuring that unnecessary background activities are not accessing computational resources throughout the solvers' performance. The unconstrained and constrained NLP problems are selected amongst the standard benchmark problems [27–29], and they are reported in [72]. Specifically, the benchmark problems include combinations of logarithmic, trigonometric, and exponential terms, non-convex and convex functions, a minimum of two to a maximum of thirty variables, and a maximum of nine constraint functions for the constrained optimization problems. As mentioned in Section 3.3, the comparison between each solver is carried out by considering three different settings: plug and play, high accuracy, and quick solution. In this way, we want to assess the robustness, accuracy, and computational time of every solver. For each benchmark problem, all solvers use the same set of randomly generated initial guesses.

### 4.1. Results for Unconstrained Optimization Problems

A collection of 30 unconstrained optimization test problems is used to validate the performance of the optimization algorithms. For the purpose of this analysis, given $N = 30$ benchmark test functions, $M = 17$ solvers and algorithms, $K = 50$ randomly generated initial guesses, and $Z = 3$ iterations, a set of $N \times M \times K \times Z$ runs are executed. Tables 13–15 report the results for the plug and play (P&P), high accuracy (HA), and quick solution (QS) settings, respectively. From the analysis of the results for the P&P settings, Table 13, we observe that all the versions of BARON have the highest convergence rate. BARON (auto) and BARON (ipopt) are able to reach the minimum mean error and variance, but they are not the fastest ones to reach the solution. Moreover, BARON (sd), BARON (sqp), SNOPT, and PENLAB are able to obtain good results in terms of mean error and variance. Overall, PENLAB is also able to reach a convergence rate similar to BARON (auto) and BARON (ipopt), with the advantage of being considerably faster than them. The worst results in terms of accuracy and convergence rate are obtained by SOLNP and SGRA. For the HA settings, Table 14, we can observe similar trends. In general, as expected, all the solvers manage to achieve a more accurate solution as they reduce the average error, increase their convergence rate, and increase the average convergence time. MIDACO is now able to reach the second highest convergence rate, after all the versions of BARON. Overall PENLAB is the solver which delivers a good trade-off in performance. With respect to the P&P settings, SOLNP significantly improves its convergence rate, whereas SGRA just slightly increase its performances. It is interesting to observe that KNITRO (sqp), aside from improving its convergence rate, increases its mean error and variance. Despite our effort, we are not sure how to explain this unexpected behaviour. Regarding the QS settings, Table 15, generally all the solvers reduce their convergence time and also decrease their convergence rate except for BARON (auto), BARON (ipopt), and BARON (sqp) which remain unaltered. SQA, FMINUNC (quasi-newton), SNOPT, and SOLNP are amongst the fastest to reach the solution but their convergence rate is quite low, except for SNOPT. In addition, conversely to all the other solvers that experience a smaller CPU time, BARON is not always able to achieve a faster CPU time with respect to the P&P settings. The same happens to the SGRA, probably due to its intrinsic iterative nature. Finally, KNITRO (sqp) delivers, in all cases, the slowest CPU time amongst the other subsolvers. This might be due to the fact that it implements, internally, Quadratic Programming subproblems characterized by computationally expensive iterations.

**Table 13.** All unconstrained problems, plug and play (P&P) settings. Solvers ranked with respect to convergence rate.

| Ranking | Solver | $\bar{E}$ [%] | $\bar{\sigma}$ [%] | $\bar{\gamma}$ [%] | $\overline{CPU}$ [s] | Free |
|---|---|---|---|---|---|---|
| 1 | BARON (auto) | $2.766 \times 10^{-6}$ | $1.448 \times 10^{-31}$ | **94.7** | 0.208 | No |
| 2 | BARON (ipopt) | $2.766 \times 10^{-6}$ | $1.443 \times 10^{-31}$ | **94.7** | 0.205 | No |
| 3 | BARON (sd) | $8.469 \times 10^{-5}$ | $1.743 \times 10^{-9}$ | **94.7** | 0.216 | No |
| 4 | BARON (sqp) | $1.690 \times 10^{-7}$ | $2.821 \times 10^{-20}$ | **94.7** | 0.195 | No |
| 5 | PENLAB | $1.016 \times 10^{-3}$ | $5.340 \times 10^{-37}$ | **88.5** | 0.0125 | Yes |
| 6 | MIDACO | $1.435 \times 10^{-1}$ | $1.445 \times 10^{-1}$ | **88.4** | 0.349 | No |
| 7 | SNOPT | $7.008 \times 10^{-3}$ | $2.444 \times 10^{-2}$ | **73.8** | 0.0071 | No |
| 8 | FMINUNC (trust-region) | $7.836 \times 10^{-2}$ | $2.068 \times 10^{-2}$ | **68.8** | 0.0153 | No |
| 9 | KNITRO (sqp) | $8.139 \times 10^{-2}$ | $9.254 \times 10^{-2}$ | **60.8** | 0.049 | No |
| 10 | KNITRO (interior-point/D) | $1.045 \times 10^{-1}$ | $1.279 \times 10^{-1}$ | **60.3** | 0.019 | No |
| 11 | KNITRO (interior-point/CG) | $1.045 \times 10^{-1}$ | $1.066 \times 10^{-1}$ | **59.9** | 0.017 | No |
| 12 | KNITRO (active-set) | $1.163 \times 10^{-1}$ | $1.409 \times 10^{-1}$ | **59.8** | 0.017 | No |
| 13 | FMINUNC (quasi-newton) | $8.643 \times 10^{-2}$ | $1.669 \times 10^{-1}$ | **52.8** | 0.0045 | No |
| 14 | SQA | $2.362 \times 10^{-1}$ | $1.383 \times 10^{-1}$ | **52.5** | 0.0005 | Yes |
| 15 | MQA | $2.031 \times 10^{-1}$ | $8.748 \times 10^{-2}$ | **51.7** | 0.1345 | Yes |
| 16 | SOLNP | $4.648 \times 10^{-1}$ | $1.908 \times 10^{-1}$ | **48.2** | 0.0097 | Yes |
| 17 | SGRA | $5.921 \times 10^{-1}$ | $8.627 \times 10^{-2}$ | **40.8** | 0.2227 | Yes |

**Table 14.** All unconstrained problems, high accuracy (HA) settings. Solvers ranked with respect to mean error.

| Ranking | Solver | $\bar{E}$ [%] | $\bar{\sigma}$ [%] | $\bar{\gamma}$ [%] | $\overline{CPU}$ [s] | Free |
|---|---|---|---|---|---|---|
| 1 | BARON (sqp) | $\mathbf{1.690} \times 10^{-7}$ | $2.821 \times 10^{-20}$ | 94.7 | 0.194 | No |
| 2 | BARON (auto) | $\mathbf{2.107} \times 10^{-7}$ | $1.448 \times 10^{-31}$ | 94.7 | 0.208 | No |
| 3 | BARON (ipopt) | $\mathbf{2.107} \times 10^{-7}$ | $1.443 \times 10^{-31}$ | 94.7 | 0.206 | No |
| 4 | BARON (sd) | $\mathbf{1.433} \times 10^{-6}$ | $2.408 \times 10^{-13}$ | 94.7 | 0.301 | No |
| 5 | PENLAB | $\mathbf{4.042} \times 10^{-6}$ | $8.944 \times 10^{-42}$ | 88.5 | 0.0121 | Yes |
| 6 | SOLNP | $\mathbf{9.420} \times 10^{-4}$ | $7.900 \times 10^{-4}$ | 69.1 | 0.0095 | Yes |
| 7 | SNOPT | $\mathbf{1.260} \times 10^{-3}$ | $1.298 \times 10^{-3}$ | 74.2 | 0.0099 | No |
| 8 | MQA | $\mathbf{3.160} \times 10^{-3}$ | $5.835 \times 10^{-5}$ | 52.1 | 0.1520 | Yes |
| 9 | FMINUNC (quasi-newton) | $\mathbf{3.526} \times 10^{-3}$ | $8.852 \times 10^{-4}$ | 59.2 | 0.0062 | No |
| 10 | SQA | $\mathbf{3.984} \times 10^{-3}$ | $9.000 \times 10^{-5}$ | 53.2 | 0.0003 | Yes |
| 11 | FMINUNC (trust-region) | $\mathbf{1.860} \times 10^{-2}$ | $1.423 \times 10^{-2}$ | 68.8 | 0.0238 | No |
| 12 | KNITRO (interior-point/D) | $\mathbf{7.130} \times 10^{-2}$ | $1.921 \times 10^{-1}$ | 67.8 | 0.021 | No |
| 13 | KNITRO (active-set) | $\mathbf{7.200} \times 10^{-2}$ | $1.883 \times 10^{-1}$ | 67.8 | 0.022 | No |
| 14 | KNITRO (interior-point/CG) | $\mathbf{7.467} \times 10^{-2}$ | $1.381 \times 10^{-1}$ | 68.0 | 0.022 | No |
| 15 | MIDACO | $\mathbf{7.754} \times 10^{-2}$ | $7.188 \times 10^{-2}$ | 92.0 | 1.035 | No |
| 16 | KNITRO (sqp) | $\mathbf{1.034} \times 10^{-1}$ | $1.785 \times 10^{-1}$ | 68.9 | 0.074 | No |
| 17 | SGRA | $\mathbf{2.709} \times 10^{-1}$ | $1.335 \times 10^{-1}$ | 44.9 | 0.2555 | Yes |

**Table 15.** All unconstrained problems, quick solution (QS) settings. Solvers ranked with respect to mean CPU time.

| Ranking | Solver | $\bar{E}$ [%] | $\bar{\sigma}$ [%] | $\bar{\gamma}$ [%] | $\overline{CPU}$ [s] | Free |
|---|---|---|---|---|---|---|
| 1 | SQA | $1.964 \times 10^{-1}$ | $1.609 \times 10^{-1}$ | 43.3 | **0.0002** | Yes |
| 2 | FMINUNC (quasi-newton) | $6.076 \times 10^{-1}$ | $8.157 \times 10^{-1}$ | 33.8 | **0.0024** | No |
| 3 | SNOPT | $1.581 \times 10^{-1}$ | $1.367 \times 10^{-1}$ | 66.4 | **0.0040** | No |
| 4 | SOLNP | $5.357 \times 10^{-1}$ | $3.847 \times 10^{-1}$ | 41.2 | **0.0093** | Yes |
| 5 | FMINUNC (trust-region) | $1.924 \times 10^{-1}$ | $1.997 \times 10^{-1}$ | 49.3 | **0.0108** | No |
| 6 | PENLAB | $5.452 \times 10^{-5}$ | $5.623 \times 10^{-39}$ | 84.6 | **0.0118** | Yes |
| 7 | KNITRO (interior-point/D) | $4.555 \times 10^{-1}$ | $4.975 \times 10^{-1}$ | 45.5 | **0.014** | No |
| 8 | KNITRO (interior-point/CG) | $4.851 \times 10^{-1}$ | $3.918 \times 10^{-1}$ | 44.2 | **0.014** | No |
| 9 | KNITRO (active-set) | $4.927 \times 10^{-1}$ | $4.737 \times 10^{-1}$ | 44.1 | **0.014** | No |
| 10 | KNITRO (sqp) | $5.821 \times 10^{-1}$ | $5.660 \times 10^{-1}$ | 46.9 | **0.022** | No |
| 11 | MIDACO | $2.455 \times 10^{-2}$ | $2.985 \times 10^{-2}$ | 74.6 | **0.070** | No |
| 12 | MQA | $2.405 \times 10^{-1}$ | $2.930 \times 10^{-1}$ | 42.2 | **0.1819** | Yes |
| 13 | BARON (sqp) | $1.690 \times 10^{-7}$ | $2.821 \times 10^{-20}$ | 94.7 | **0.198** | No |
| 14 | BARON (ipopt) | $2.602 \times 10^{-6}$ | $1.443 \times 10^{-31}$ | 94.7 | **0.204** | No |
| 15 | BARON (auto) | $2.602 \times 10^{-6}$ | $1.448 \times 10^{-31}$ | 94.7 | **0.205** | No |
| 16 | BARON (sd) | $3.929 \times 10^{-6}$ | $1.846 \times 10^{-9}$ | 89.5 | **0.206** | No |
| 17 | SGRA | $8.640 \times 10^{-1}$ | $2.211 \times 10^{-1}$ | 23.8 | **0.3033** | Yes |

*4.2. Results for Constrained Optimization Problems*

A collection of 30 constrained optimization test problems is used to validate the performance of the optimization algorithms. For the purpose of the analysis, given $N = 30$ benchmark test functions, $M = 22$ solvers and algorithms, $K = 50$ randomly generated initial guesses, and $Z = 3$ iterations, a set of $N \times M \times K \times Z$ runs are executed. Tables 16–18 report the results for the P&P, HA, and QS settings, respectively. From the analysis of the results for the P&P settings, Table 16, we observe that all the versions of BARON are able to reach the highest accuracy and the best convergence rate but they are not the fastest to reach the solution. KNITRO (interior-point/D) is able to achieve the second best convergence rate, with an average CPU time that is two order of magnitude faster than BARON. PENLAB obtains the best mean error and variance but this performance is tempered by a low convergence rate, together with the SGRA, MQA, and SQA which are also quite slow to reach a solution. SNOPT reaches a convergence rate slightly lower than KNITRO (interior-point/D), KNITRO (interior-point/CG), FMINCON (interior-point), and KNITRO (sqp) but is significantly faster. Regarding the HA settings, Table 17, similar consideration can be made for BARON, but in this case the CPU time is increasing. MIDACO shows an improvement in the convergence rate, reaching values very similar to BARON. PENLAB still obtains the best mean error and variance, but it has one of the lowest convergence rates, together with the SGRA. In general, most of the solvers increase their convergence rate, and decrease their mean error, except for GCMMA and PENLAB. Regarding the QS settings, Table 18, generally all the solvers decrease their convergence rate except for BARON and PENLAB. The same considerations about BARON and PENLAB can be done as in the two previous scenarios. MIDACO reports a significant decrease in the convergence rate. SNOPT, FMINCON, and KNITRO algorithms reach a convergence rate lower than BARON, but not as low as other solvers (SONLP, APSO, PENLAB), and they are significantly faster. Also for the constrained problems, KNITRO (sqp) delivers the slowest CPU time amongst the other subsolvers. Again, this might be a consequence of the computationally heavy internal Quadratic Programming subproblems. The worst results in terms of convergence rate and CPU time are obtained by MQA and SQA.

**Table 16.** All constrained problems, plug and play (P&P) settings. Solvers ranked with respect to convergence rate.

| Ranking | Solver | $\bar{E}$ [%] | $\bar{\sigma}$ [%] | $\bar{\gamma}$ [%] | $\overline{CPU}$ [s] | Free |
|---|---|---|---|---|---|---|
| 1 | BARON (auto) | $1.299 \times 10^{-1}$ | $2.141 \times 10^{-7}$ | **92.0** | 1.497 | No |
| 2 | BARON (ipopt) | $1.298 \times 10^{-1}$ | $2.894 \times 10^{-7}$ | **92.0** | 1.767 | No |
| 3 | BARON (sd) | $1.296 \times 10^{-1}$ | $5.178 \times 10^{-7}$ | **92.0** | 1.379 | No |
| 4 | BARON (sqp) | $1.298 \times 10^{-1}$ | $2.332 \times 10^{-7}$ | **92.0** | 1.412 | No |
| 5 | KNITRO (interior-point/D) | $1.782 \times 10^{-1}$ | $2.035 \times 10^{-1}$ | **77.7** | 0.034 | No |
| 6 | KNITRO (interior-point/CG) | $1.756 \times 10^{-1}$ | $2.085 \times 10^{-1}$ | **77.5** | 0.033 | No |
| 7 | FMINCON (interior-point) | $1.985 \times 10^{-1}$ | $2.413 \times 10^{-1}$ | **75.9** | 0.0271 | No |
| 8 | KNITRO (sqp) | $1.851 \times 10^{-1}$ | $2.102 \times 10^{-1}$ | **75.7** | 0.160 | No |
| 9 | SNOPT | $1.689 \times 10^{-1}$ | $2.010 \times 10^{-1}$ | **72.1** | 0.0040 | No |
| 10 | FMINCON (active-set) | $1.795 \times 10^{-1}$ | $2.123 \times 10^{-1}$ | **71.9** | 0.0204 | No |
| 11 | FMINCON (sqp-legacy) | $1.893 \times 10^{-1}$ | $2.429 \times 10^{-1}$ | **69.4** | 0.0111 | No |
| 12 | KNITRO (active-set) | $1.994 \times 10^{-1}$ | $2.702 \times 10^{-1}$ | **72.2** | 0.070 | No |
| 13 | FMINCON (sqp) | $1.908 \times 10^{-1}$ | $2.446 \times 10^{-1}$ | **69.3** | 0.0093 | No |
| 14 | MIDACO | $7.348 \times 10^{-1}$ | $3.500 \times 10^{-1}$ | **66.9** | 0.353 | No |
| 15 | SOLNP | $3.243 \times 10^{-1}$ | $3.211 \times 10^{-1}$ | **48.1** | 0.0095 | Yes |
| 16 | GCMMA | $4.490 \times 10^{-1}$ | $3.742 \times 10^{-1}$ | **45.7** | 0.9681 | Yes |
| 17 | MMA | $7.188 \times 10^{-1}$ | $5.743 \times 10^{-1}$ | **44.1** | 0.5856 | Yes |
| 18 | APSO | 1.512 | 1.025 | **39.2** | 0.1772 | Yes |
| 19 | PENLAB | $1.127 \times 10^{-4}$ | $3.258 \times 10^{-41}$ | **31.0** | 0.0379 | Yes |
| 20 | SGRA | $6.360 \times 10^{-1}$ | $7.011 \times 10^{-1}$ | **30.3** | 0.9815 | Yes |
| 21 | MQA | $5.125 \times 10^{-1}$ | $3.460 \times 10^{-1}$ | **20.8** | 3.1559 | Yes |
| 22 | SQA | $3.990 \times 10^{-1}$ | $5.778 \times 10^{-1}$ | **20.2** | 3.1822 | Yes |

**Table 17.** All constrained problems, high accuracy (HA) settings. Solvers ranked with respect to mean error.

| Ranking | Solver | $\bar{E}$ [%] | $\bar{\sigma}$ [%] | $\bar{\gamma}$ [%] | $\overline{CPU}$ [s] | Free |
|---|---|---|---|---|---|---|
| 1 | PENLAB | $\mathbf{1.502} \times 10^{-4}$ | $6.711 \times 10^{-39}$ | 31.0 | 0.0488 | Yes |
| 2 | BARON (auto) | $\mathbf{1.943} \times 10^{-3}$ | $5.777 \times 10^{-18}$ | 92.0 | 2.056 | No |
| 3 | BARON (sd) | $\mathbf{1.943} \times 10^{-3}$ | $1.405 \times 10^{-16}$ | 92.0 | 2.469 | No |
| 4 | BARON (sqp) | $\mathbf{1.943} \times 10^{-3}$ | $4.745 \times 10^{-17}$ | 92.0 | 2.041 | No |
| 5 | BARON (ipopt) | $\mathbf{8.074} \times 10^{-2}$ | $9.457 \times 10^{-10}$ | 90.8 | 3.568 | No |
| 6 | SNOPT | $\mathbf{1.689} \times 10^{-1}$ | $2.010 \times 10^{-1}$ | 72.4 | 0.0069 | No |
| 7 | KNITRO (interior-point/CG) | $\mathbf{1.754} \times 10^{-1}$ | $2.085 \times 10^{-1}$ | 77.7 | 0.040 | No |
| 8 | FMINCON (active-set) | $\mathbf{1.770} \times 10^{-1}$ | $2.082 \times 10^{-1}$ | 72.3 | 0.0214 | No |
| 9 | KNITRO (interior-point/D) | $\mathbf{1.782} \times 10^{-1}$ | $2.035 \times 10^{-1}$ | 78.0 | 0.039 | No |
| 10 | FMINCON (sqp-legacy) | $\mathbf{1.857} \times 10^{-1}$ | $2.365 \times 10^{-1}$ | 69.7 | 0.0110 | No |
| 11 | KNITRO (sqp) | $\mathbf{1.872} \times 10^{-1}$ | $2.120 \times 10^{-1}$ | 75.6 | 0.209 | No |
| 12 | FMINCON (sqp) | $\mathbf{1.881} \times 10^{-1}$ | $2.388 \times 10^{-1}$ | 69.4 | 0.0082 | No |
| 13 | KNITRO (active-set) | $\mathbf{1.951} \times 10^{-1}$ | $2.791 \times 10^{-1}$ | 73.4 | 0.078 | No |
| 14 | FMINCON (interior-point) | $\mathbf{1.985} \times 10^{-1}$ | $2.413 \times 10^{-1}$ | 75.9 | 0.0326 | No |
| 15 | SQA | $\mathbf{2.754} \times 10^{-1}$ | $2.838 \times 10^{-1}$ | 20.1 | 3.1838 | Yes |
| 16 | SOLNP | $\mathbf{2.949} \times 10^{-1}$ | $3.106 \times 10^{-1}$ | 44.8 | 0.0112 | Yes |
| 17 | MIDACO | $\mathbf{3.662} \times 10^{-1}$ | $2.938 \times 10^{-1}$ | 87.0 | 1.053 | No |
| 18 | GCMMA | $\mathbf{5.112} \times 10^{-1}$ | $5.668 \times 10^{-1}$ | 45.2 | 1.0599 | Yes |
| 19 | MQA | $\mathbf{5.358} \times 10^{-1}$ | $4.601 \times 10^{-1}$ | 20.8 | 3.2012 | Yes |
| 20 | SGRA | $\mathbf{6.248} \times 10^{-1}$ | $7.673 \times 10^{-1}$ | 30.0 | 0.9632 | Yes |
| 21 | MMA | $\mathbf{9.786} \times 10^{-1}$ | $5.748 \times 10^{-1}$ | 42.1 | 0.7101 | Yes |
| 22 | APSO | **1.173** | 1.014 | 45.9 | 1.0168 | Yes |

**Table 18.** All constrained problems, quick solution (QS) settings. Solvers ranked with respect to mean CPU time.

| Ranking | Solver | $\bar{E}$ [%] | $\bar{\sigma}$ [%] | $\bar{\gamma}$ [%] | $\overline{CPU}$ [s] | Free |
|---|---|---|---|---|---|---|
| 1 | SNOPT | $1.767 \times 10^{-1}$ | $2.045 \times 10^{-1}$ | 70.2 | **0.0027** | No |
| 2 | FMINCON (sqp) | $1.916 \times 10^{-1}$ | $2.448 \times 10^{-1}$ | 69.0 | **0.0071** | No |
| 3 | SOLNP | $4.790 \times 10^{-1}$ | $6.452 \times 10^{-1}$ | 46.6 | **0.0087** | Yes |
| 4 | FMINCON (sqp-legacy) | $1.902 \times 10^{-1}$ | $2.431 \times 10^{-1}$ | 69.1 | **0.0092** | No |
| 5 | FMINCON (active-set) | $2.850 \times 10^{-1}$ | $3.484 \times 10^{-1}$ | 68.9 | **0.0165** | No |
| 6 | KNITRO (interior-point/D) | $2.221 \times 10^{-1}$ | $2.700 \times 10^{-1}$ | 72.7 | **0.024** | No |
| 7 | FMINCON (interior-point) | $2.166 \times 10^{-1}$ | $2.554 \times 10^{-1}$ | 72.3 | **0.0262** | No |
| 8 | KNITRO (interior-point/CG) | $3.082 \times 10^{-1}$ | $3.585 \times 10^{-1}$ | 72.2 | **0.028** | No |
| 9 | KNITRO (active-set) | $2.212 \times 10^{-1}$ | $2.975 \times 10^{-1}$ | 69.3 | **0.030** | No |
| 10 | PENLAB | $1.896 \times 10^{-4}$ | $1.454 \times 10^{-37}$ | 31.0 | **0.0323** | Yes |
| 11 | APSO | 1.531 | $5.677 \times 10^{-1}$ | 35.2 | **0.0538** | Yes |
| 12 | KNITRO (sqp) | $2.458 \times 10^{-1}$ | $3.221 \times 10^{-1}$ | 72.5 | **0.063** | No |
| 13 | MIDACO | $9.737 \times 10^{-1}$ | $8.263 \times 10^{-1}$ | 53.0 | **0.070** | No |
| 14 | SGRA | $8.774 \times 10^{-1}$ | 1.198 | 27.5 | **0.9369** | Yes |
| 15 | MMA | 1.161 | 1.189 | 41.0 | **0.1324** | Yes |
| 16 | GCMMA | $6.967 \times 10^{-1}$ | $4.256 \times 10^{-1}$ | 45.5 | **0.5574** | Yes |
| 17 | BARON (sqp) | $1.395 \times 10^{-1}$ | $8.212 \times 10^{-5}$ | 92.0 | **0.869** | No |
| 18 | BARON (auto) | $1.374 \times 10^{-1}$ | $4.647 \times 10^{-5}$ | 92.0 | **0.874** | No |
| 19 | BARON (ipopt) | $1.369 \times 10^{-1}$ | $2.163 \times 10^{-5}$ | 92.0 | **0.880** | No |
| 20 | BARON (sd) | $1.370 \times 10^{-1}$ | $1.291 \times 10^{-5}$ | 92.0 | **0.999** | No |
| 21 | MQA | $5.844 \times 10^{-1}$ | $4.193 \times 10^{-1}$ | 20.8 | **3.1174** | Yes |
| 22 | SQA | $3.316 \times 10^{-1}$ | $3.131 \times 10^{-1}$ | 20.1 | **3.1361** | Yes |

## 5. UAV Path Planning: Real-World Application Benchmark

Path planning poses a significant challenge for autonomous UAVs, especially when specific mission criteria must be fulfilled. In general, path planning problems can be formulated as optimal control problems. i.e., letting the states trajectories and control inputs of the vehicles be denoted by $\boldsymbol{x}(t)$ and $\boldsymbol{u}(t)$, respectively; the path planning problem can formally be stated as follows:

$$\min_{\boldsymbol{x}(t), \boldsymbol{u}(t)} I(\boldsymbol{x}(t), \boldsymbol{u}(t)) \tag{20}$$

subject to

$$\dot{\boldsymbol{x}}(t) = \boldsymbol{f}(\boldsymbol{x}(t), \boldsymbol{u}(t)), \quad \forall t \in [0, t_f], \tag{21}$$

$$\boldsymbol{e}(\boldsymbol{x}(0), \boldsymbol{x}(t_f)) = \boldsymbol{0}, \tag{22}$$

$$\boldsymbol{h}(\boldsymbol{x}(t), \boldsymbol{u}(t)) \leq \boldsymbol{0}, \quad \forall t \in [0, t_f], \tag{23}$$

where $I : \mathbb{R}^{n_x} \times \mathbb{R}^{n_u} \to \mathbb{R}$ is a cost function, $\boldsymbol{f} : \mathbb{R}^{n_x} \times \mathbb{R}^{n_u} \to \mathbb{R}^{n_x}$ represents the vehicles dynamics, $\boldsymbol{e} : \mathbb{R}^{n_x} \times \mathbb{R}^{n_x} \to \mathbb{R}^{n_e}$ and $\boldsymbol{h} : \mathbb{R}^{n_x} \times \mathbb{R}^{n_u} \to \mathbb{R}^{n_h}$ are constraints.

One common criterion is optimizing the path for either minimal energy consumption or minimum time of arrival at the destination. The constraint in Equation (22) enforces the boundary conditions, e.g., initial and final position, speed, and heading angles of the vehicles, and Equation (23) describes feasibility and mission specific constraints, e.g., minimum and maximum speed, acceleration, and collision avoidance constraints. With these considerations in mind, we present a path planning example that caters to both single and multiple rotorcraft UAV scenarios. In particular, this work considers two problems, namely, a single drone landing mission, and a multiple-drone-formation flight. Both problems consider 3D trajectories, and the chosen optimization criteria is mission time. The generation of the 3D trajectories is based on Bernstein polynomial approximations of vehicles' trajectories to transcribe the infinite dimensional path planning problems into

NLP problems [30,73]. In turn, the trajectory of vehicle *i* is parameterized by the *no*-th order Bernstein polynomials

$$x(t) = \sum_{i=0}^{no} \bar{x}_i b_{i,no}(t), \qquad t \in [0, t_f], \tag{24}$$

where $\bar{x}_i$, $i = 0, \ldots, no$, are Bernstein polynomial coefficients and $b_{i,no}(t)$, $i = 0, \ldots, no$, are Bernstein polynomial basis functions of order. The following properties of Bernstein polynomials are used in this paper:

(1) *Differentiation and integration*: the $\ell$-th derivative, with $\ell \in \mathbb{N}$, of the Bernstein polynomial $x(t)$ defined above is computed as

$$\frac{d^\ell x_k(t)}{dt^\ell} = \sum_{i=0}^{no} \sum_{j=0}^{no} \bar{x}_j D_{j,i}^\ell b_{i,no}(t), \tag{25}$$

where $D_{j,i}^\ell$ is the $(j, i)$-th entry of a square differentiation matrix [74].

(2) *Arithmetic operations*: the sum (difference) of two *no*-th order Bernstein polynomials is an *no*-th order Bernstein polynomial. The product between two Bernstein polynomials of orders $no_1$ and $no_2$ is a Bernstein polynomial of order $no_1 + no_2$ [75] (Chapter 5).

Using these properties, the following functions can be expressed as Bernstein polynomials:

$$\text{speed:} \quad \|\dot{x}(t)\|^2 = \sum_{j=0}^{2no} \bar{v}_j b_{j,2no}(t),$$

$$\text{acceleration:} \quad \|\ddot{x}(t)\|^2 = \sum_{j=0}^{2no} \bar{a}_j b_{j,2no}(t), \tag{26}$$

$$\text{distance:} \quad \|x(t) - \hat{x}\|^2 = \sum_{j=0}^{2no} \bar{d}_j b_{j,2no}(t).$$

In the equation above, $\bar{v}_j$, $\bar{a}_j$ and $\bar{d}_j$, $\forall j \in \{0, \ldots, 2N\}$, can be obtained from algebraic manipulation of the Bernstein coefficients $\bar{x}_0, \ldots, \bar{x}_{no}$. With this setup, the planning problem amounts at finding optimal Bernstein polynomial coefficients such that trajectory $x(t)$ minimizes some objective function and satisfies a set of constraints.

This benchmark comparison is performed considering the solvers' plug and play (P&P) setting. Every test problem has been initially evaluated by using FMINCON (sqp) with ad hoc settings, as reported in Table 19. These settings are characterized by very stringent tolerances and a high number of maximum iterations. This provide us with the candidate optimal solution $f(x^*)$ for each randomly generated initial guess. The setting $exitflag = 1$ means that the first-order optimality measure is less than *OptimalityTolerance*, and the maximum constraint violation is less than *ConstraintTolerance*. It must be noted that only for the evaluation of the candidate optimal solution referring to the UAV formation flying problem with $nv = 25$ (see Section 5.2), the tolerances have been relaxed to $10^{-6}$. This allow us to adopt the convergence metrics defined in Section 3.2. In turn, the convergence rate is computed considering successful a run that satisfies the converging threshold conditions $E_k \leq E_{max} = 5\%$, and $CPU_k \leq CPU_{max}$ for every test problem (see Tables 20 and 24), for the run starting from the $k$-th initial guess. The $CPU_{max}$ are chosen by adequately increasing the CPU time required by the ad hoc settings to solve each problem.

**Table 19.** FMINCON (sqp) ad hoc settings.

| Settings | Ad hoc |
|---|---|
| *exit flag* | 1 |
| *MaxFunctionEvaluations* | 3,000,000 |
| *MaxIterations* | 100,000 |
| *StepTolerance* | $10^{-14}$ |
| *ConstraintTolerance* | $10^{-14}$ |
| *FunctionTolerance* | $10^{-14}$ |
| *OptimalityTolerance* | $10^{-14}$ |

*5.1. The 3D Minimum Time Problem: UAV Landing*

The first problem is a minimum time problem for a simplified 3D model of a multi-rotor drone. The vehicle is required to reach the origin in minimum time from a given initial condition with all the control inputs bounded by $\pm 1$. The problem is defined as follows:

$$\min_{x,u,t_f} J = t_f, \tag{27}$$

subject to

$$
\begin{cases}
\dot{x}_1 = x_4, \quad \dot{x}_2 = x_5, \quad \dot{x}_3 = x_6, \\
\dot{x}_4 = u_1, \quad \dot{x}_5 = u_2, \quad \dot{x}_6 = -g + u_3, \\
x_1(0) = k_1 \quad x_2(0) = k_2, \quad x_3(0) = k_3, \\
x_4(0) = k_4, \quad x_5(0) = k_5, \quad x_6(0) = k_6, \\
x_1(t_f) = x_2(t_f) = x_3(t_f) = 0, \\
x_4(t_f) = x_5(t_f) = x_6(t_f) = 0, \\
|u_1(t)| \le 1, \quad |u_2(t)| \le 1, \quad |-g + u_3(t)| \le 1, \quad \forall t \in [0, t_f],
\end{cases}
\tag{28}
$$

where $x_1, x_2, x_3, x_4, x_5, x_6$ are the Bernstein polynomials associated to the drone position and velocity, $u_1, u_2, u_3$ are Bernstein polynomials associated to the control input, $g$ is the gravitational acceleration, $k_{1...6}$ are the randomly generated initial conditions, and $t_f$ is the final maneuver time to be minimized. The Bernstein polynomials are vectors of dimension *no+1*, being *no* the polynomial order of approximation. More accurate results can be obtained for larger *no*, at the expense of the computational time. The dimension of the problem is $9 \times (no + 1) + 1$, with the last 1 being the final time $t_f$, with $6 \times (no + 1) + 12$ equality constraints, and $6 \times (no + 1)$ inequality constraints. The derivative is computed using the squared differentiation matrix for Bernstein polynomials [30]. Figure 1 reports an example of solution for the 3D minimum time UAV landing problem with $no = 50$. Additional details regarding the problem can be found in [35].

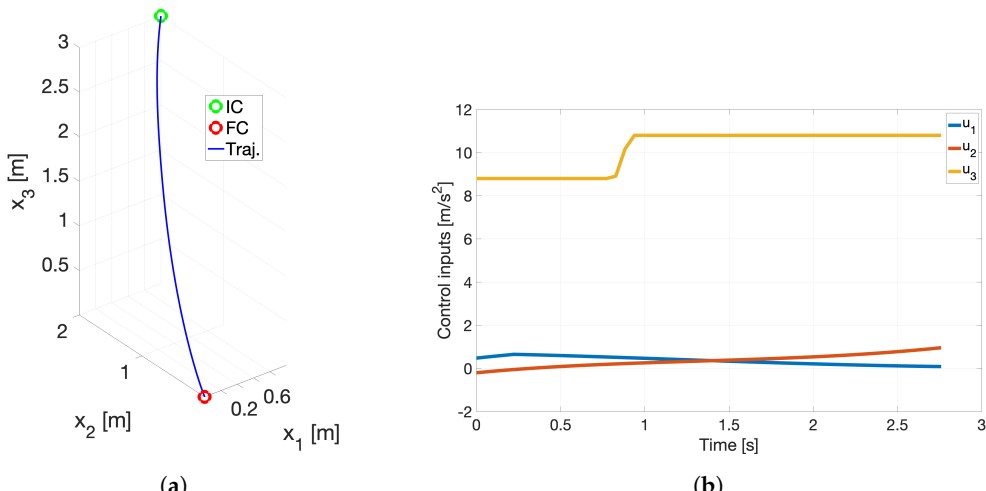

(a)    (b)

**Figure 1.** Example of solution for UAV Landing problem. (**a**) UAV position. (**b**) UAV control inputs.

In our analysis, given $N = 3$ benchmark test problems, each of them characterized by a different order of approximation $no$, $M = 22$ solvers and algorithms, $K = 50$ randomly generated initial guesses, and $Z = 3$ iterations, a set of $M \times K \times Z$ runs are executed for each test problem. Table 20 reports the number of variables, equality and inequality constraints for the different order of approximation $no$. Due to the increased complexity of the problems with higher values of $no$, and the way some solvers are implemented, not all the solvers are used for every benchmark test problem. In particular, for $no = 50, 150$, MQA and SQA are omitted, as they require to evaluate additional slack variables for every inequality constraints, and the Hessian of the constraint functions, which needs to be loaded from stored MATLAB binary files at each iteration. We have noticed that this is extremely computationally expensive, going over the $CPU_{max}$ limit. For $no = 50, 150$, PENLAB is omitted because the Hessians of the constraint functions need to be loaded from stored MATLAB binary files at each iteration, whereas SGRA is left out for $no = 150$ because of the use of slack variables. In both cases, this is overly computationally expensive.

**Table 20.** UAV Landing: problem dimensions, and max CPU time threshold.

| $no$ | 5 | 50 | 150 |
|---|---|---|---|
| Variables | 48 | 460 | 1360 |
| Equality constr. | 36 | 318 | 918 |
| Inequality const. | 55 | 306 | 906 |
| $CPU_{max}$ [s] | 20 | 75 | 500 |

Table 21 reports the results for the UAV landing problem with $no = 5$. It can be seen that BARON (auto), BARON (ipopt), BARON (sqp), FMINCON (sqp), FMINCON (active-set), SNOPT, KNITRO (active-set), and KNITRO (sqp) reach 100% convergence rate and similar accuracy in terms of mean error and variance. The BARON algorithms have the highest CPU time, together with the SGRA. APSO, MMA/GCMMA, MIDACO, MQA and SQA are not able to find any successful solutions that satisfy the converging threshold conditions in terms of maximum error $E_{max}$. It is interesting to note instead that PENLAB reaches a 100% convergence rate, but without satisfying the maximum CPU time. This shows how the need to load the Hessian of the constraint functions from stored MATLAB binary files at each iteration affects the computational time of PENLAB. From the UAV landing problem solutions with $no = 50$ reported in Table 22, the results show that fewer solvers are able to reach full convergence, specifically BARON (auto), BARON (ipot), BARON (sqp), KNITRO (active-set), and SNOPT, with SNOPT being the fastest solver. In this case, an additional solver is not able to satisfy the converging threshold conditions, namely, FMINCON (interior-point). Regarding the UAV landing problem solutions with $no = 150$ reported in Table 23, only BARON (auto), BARON (ipopt), and SNOPT reach 100% of convergence rate, whereas FMINCON (sqp-legacy) is not able to converge to any acceptable solutions, together with the same solvers mentioned in the previous cases. As expected, the average computational time is increasing for all the solvers, but SNOPT is able to outperform all the other solvers with a CPU time of one or two orders of magnitude less than them. The fact that KNITRO and FMINCON are no more able to reach full convergence and, at the same time, increase their CPU time is probably due to the general increase in the problem complexity (i.e., higher number of variables and constraints), and to the absence of a warm-start, since the initial conditions are randomly generated.

**Table 21.** UAV landing problem solution for $no = 5$.

| Ranking | Solver | $\bar{E}$ [%] | $\bar{\sigma}$ [%] | $\bar{\gamma}$ [%] | $\overline{CPU}$ [s] |
|---|---|---|---|---|---|
| 1 | BARON (auto) | $4.076 \times 10^{-6}$ | $7.082 \times 10^{-12}$ | **100.0** | 0.499 |
| 2 | BARON (ipopt) | $4.076 \times 10^{-6}$ | $7.084 \times 10^{-12}$ | **100.0** | 0.475 |
| 3 | BARON (sqp) | $4.084 \times 10^{-6}$ | $7.084 \times 10^{-12}$ | **100.0** | 0.359 |
| 4 | FMINCON (sqp) | $4.084 \times 10^{-6}$ | $7.084 \times 10^{-12}$ | **100.0** | 0.061 |
| 5 | FMINCON (active-set) | $3.957 \times 10^{-6}$ | $7.372 \times 10^{-12}$ | **100.0** | 0.059 |
| 6 | SNOPT | $4.101 \times 10^{-6}$ | $7.348 \times 10^{-12}$ | **100.0** | 0.023 |

**Table 21.** *Cont.*

| Ranking | Solver | $\bar{E}$ [%] | $\bar{\sigma}$ [%] | $\bar{\gamma}$ [%] | $\overline{CPU}$ [s] |
|---|---|---|---|---|---|
| 7 | KNITRO (active-set) | $4.084 \times 10^{-6}$ | $7.084 \times 10^{-12}$ | **100.0** | 0.072 |
| 8 | KNITRO (sqp) | $4.084 \times 10^{-6}$ | $7.083 \times 10^{-12}$ | **100.0** | 0.178 |
| 9 | KNITRO (interior-point/CG) | $2.144 \times 10^{-5}$ | $2.125 \times 10^{-10}$ | **98.0** | 0.145 |
| 10 | FMINCON (sqp-legacy) | $3.900 \times 10^{-6}$ | $6.299 \times 10^{-12}$ | **96.0** | 0.082 |
| 11 | BARON (sd) | $4.076 \times 10^{-1}$ | 1.144 | **78.0** | 0.920 |
| 12 | SOLNP | $4.015 \times 10^{-4}$ | $8.380 \times 10^{-7}$ | **76.0** | 0.073 |
| 13 | SGRA | $1.908 \times 10^{-4}$ | $5.960 \times 10^{-8}$ | **46.0** | 0.804 |
| 14 | KNITRO (interior-point/D) | $4.573 \times 10^{-6}$ | $9.251 \times 10^{-12}$ | **42.0** | 0.234 |
| 15 | FMINCON (interior-point) | $6.862 \times 10^{-1}$ | 1.392 | **26.0** | 0.142 |
| - | APSO | $>E_{max}$ | - | **-** | - |
| - | GCMMA | $>E_{max}$ | - | **-** | - |
| - | MIDACO | $>E_{max}$ | - | **-** | - |
| - | MMA | $>E_{max}$ | - | **-** | - |
| - | MQA | $>E_{max}$ | - | **-** | - |
| - | PENLAB | $3.863 \times 10^{-6}$ | $8.382 \times 10^{-12}$ | **100.0** | $> CPU_{max}$ |
| - | SQA | $>E_{max}$ | - | **-** | - |

**Table 22.** UAV landing problem solution for $no = 50$.

| Ranking | Solver | $\bar{E}$ [%] | $\bar{\sigma}$ [%] | $\bar{\gamma}$ [%] | $\overline{CPU}$ [s] |
|---|---|---|---|---|---|
| 1 | BARON (auto) | $4.148 \times 10^{-6}$ | $9.990 \times 10^{-12}$ | **100.0** | 5.802 |
| 2 | BARON (ipopt) | $4.148 \times 10^{-6}$ | $9.990 \times 10^{-12}$ | **100.0** | 5.647 |
| 3 | BARON (sqp) | $4.141 \times 10^{-6}$ | $9.977 \times 10^{-12}$ | **100.0** | 5.115 |
| 4 | KNITRO (active-set) | $4.141 \times 10^{-6}$ | $9.977 \times 10^{-12}$ | **100.0** | 5.763 |
| 5 | SNOPT | $4.101 \times 10^{-6}$ | $1.000 \times 10^{-11}$ | **100.0** | 1.001 |
| 6 | KNITRO (sqp) | $4.125 \times 10^{-6}$ | $1.017 \times 10^{-11}$ | **98.0** | 16.402 |
| 7 | FMINCON (active-set) | $3.155 \times 10^{-4}$ | $3.774 \times 10^{-7}$ | **92.0** | 26.765 |
| 8 | KNITRO (interior-point/CG) | $9.074 \times 10^{-2}$ | $1.511 \times 10^{-1}$ | **84.0** | 46.202 |
| 9 | KNITRO (interior-point/D) | $1.351 \times 10^{-1}$ | $6.208 \times 10^{-1}$ | **68.0** | 23.080 |
| 10 | BARON (sd) | $6.585 \times 10^{-1}$ | 2.221 | **32.0** | 48.799 |
| 11 | FMINCON (sqp-legacy) | $4.991 \times 10^{-6}$ | $1.371 \times 10^{-11}$ | **58.0** | 24.868 |
| 12 | FMINCON (sqp) | $4.973 \times 10^{-6}$ | $1.325 \times 10^{-11}$ | **60.0** | 19.498 |
| 13 | SOLNP | $1.424 \times 10^{-2}$ | $4.673 \times 10^{-4}$ | **44.0** | 4.863 |
| - | APSO | $>E_{max}$ | - | **-** | - |
| - | FMINCON (interior-point) | $>E_{max}$ | - | **-** | - |
| - | GCMMA | $>E_{max}$ | - | **-** | - |
| - | MIDACO | $>E_{max}$ | - | **-** | - |
| - | MMA | $>E_{max}$ | - | **-** | - |
| - | SGRA | $>E_{max}$ | - | **-** | - |

**Table 23.** UAV landing problem solution for $no = 150$.

| Ranking | Solver | $\bar{E}$ [%] | $\bar{\sigma}$ [%] | $\bar{\gamma}$ [%] | $\overline{CPU}$ [s] |
|---|---|---|---|---|---|
| 1 | BARON (auto) | $3.679 \times 10^{-6}$ | $7.324 \times 10^{-12}$ | **100.0** | 70.685 |
| 2 | BARON (ipopt) | $3.678 \times 10^{-6}$ | $7.352 \times 10^{-12}$ | **100.0** | 70.194 |
| 3 | SNOPT | $3.766 \times 10^{-6}$ | $6.505 \times 10^{-12}$ | **100.0** | 7.396 |
| 4 | KNITRO (sqp) | $8.930 \times 10^{-2}$ | $2.950 \times 10^{-1}$ | **74.0** | 211.029 |
| 5 | KNITRO (active-set) | $6.241 \times 10^{-2}$ | $1.285 \times 10^{-1}$ | **66.0** | 186.258 |
| 6 | KNITRO (interior-point/D) | $4.840 \times 10^{-1}$ | 1.237 | **62.0** | 352.009 |
| 7 | SOLNP | $1.141 \times 10^{-1}$ | $1.665 \times 10^{-3}$ | **18.0** | 178.791 |
| 8 | BARON (sd) | 1.558 | 3.421 | **12.0** | 424.121 |
| 9 | FMINCON (sqp) | $3.041 \times 10^{-3}$ | $2.228 \times 10^{-5}$ | **12.0** | 411.148 |
| 10 | BARON (sqp) | 1.870 | 3.548 | **10.0** | 430.064 |
| 11 | FMINCON (active-set) | $1.196 \times 10^{-2}$ | $3.600 \times 10^{-4}$ | **8.0** | 386.340 |
| 12 | KNITRO (interior-point/CG) | $4.848 \times 10^{-4}$ | $4.374 \times 10^{-7}$ | **6.0** | 371.203 |
| - | APSO | $>E_{max}$ | - | **-** | - |
| - | FMINCON (interior-point) | $>E_{max}$ | - | **-** | - |
| - | FMINCON (sqp-legacy) | $>E_{max}$ | - | **-** | - |
| - | GCMMA | $>E_{max}$ | - | **-** | - |
| - | MIDACO | $>E_{max}$ | - | **-** | - |
| - | MMA | $>E_{max}$ | - | **-** | - |

*5.2. The 3D Minimum Time Problem: UAV Formation Flying*

The second problem is a minimum time UAV formation flying problem. Starting from a grid at altitude $z_{alt_{in}} = 0$, i.e., the initial condition, $nv$ drones need to form a circle-shaped formation at $z_{alt_f} = 10$, i.e., the final condition. The trajectories must satisfy zero speed at arrival, minimum and maximum acceleration rates, and collision avoidance constraints. The vehicles need to arrive at destination at the same final time $t_f$. As in the previous test case, $no$ is the Bernstein polynomial order of approximation, representing the number of nodes for each trajectory, whereas, $nv$ is the number of vehicles involved in the mission. Because the complexity can grow up quite quickly, a fixed value of nodes is selected, $no = 4$. The initial and final conditions are randomly generated by varying the initial grid size, the final radius of the circle-shaped formation, and the final position on the circle for each drone. The control inputs are bounded by $\pm1$, and the collision avoidance constraints are implemented to maintain a minimum distance greater or equal than 0.1 between the drones. The problem is defined as follows:

$$\min_{\substack{x_1,\dots,x_{nv} \\ u_1,\dots,u_{nv} \\ t_f}} J = t_f, \tag{29}$$

subject to

$$\begin{cases} \dot{x}_{j,1} = x_{j,4}, \quad \dot{x}_{j,2} = x_{j,5}, \quad \dot{x}_{j,3} = x_{j,6}, \\ \dot{x}_{j,4} = u_{j,1}, \quad \dot{x}_{j,5} = u_{j,2}, \quad \dot{x}_{j,6} = -g + u_{j,3}, \\ x_{j,1}(0) = x_{grid_j}, \quad x_{j,2}(0) = y_{grid_j}, \quad x_{j,3}(0) = z_{alt_{in}}, \\ x_{j,4}(0) = x_{j,5}(0) = x_{j,6}(0) = 0, \\ x_{j,1}(t_f) = x_{circle_j}, \quad x_{j,2}(t_f) = y_{circle_j}, \quad x_{j,3}(t_f) = z_{alt_f}, \\ x_{j,4}(t_f) = x_{j,5}(t_f) = x_{j,6}(t_f) = 0, \\ |u_{j,1}(t)| \le 1, \quad |u_{j,2}(t)| \le 1, \quad |-g + u_{j,3}(t)| \le 1, \quad \forall t \in [0, t_f], \\ \|p_j(t) - p_i(t)\| \ge d_{\text{safe}}, \quad \forall t \in [0, t_f], \end{cases} \tag{30}$$

for all $j, i = 1, \dots, nv$ where $nv$ is the number of vehicles involved in the mission. In the problem above, $x_{j,1}, x_{j,2}, x_{j,3}, x_{j,4}, x_{j,5}, x_{j,6}$ are the Bernstein polynomials associated to the $j$-th multi-rotor drone position and velocity, $u_{j,1}, u_{j,2}, u_{j,3}$ are the Bernstein polynomials associated to its control input, $g$ is the gravitational acceleration, $t_f$ is the final maneuver time to be minimized and shared by all the vehicles, and $x_{grid}, y_{grid}$ and $x_{circle}, y_{circle}$ are random vectors in $\mathbb{R}^3$, representing random points on a grid and a circle, respectively. In other words, the vehicles start from a grid formation and eventually converge into a circle formation. The Bernstein polynomials are vectors of dimension $no+1$, being $no$ the polynomial order of approximation. The dimension of the problem is $nv \times 9 \times (no + 1) + 1$, with the last 1 being the final time $t_f$, with $nv \times (6 \times (no + 1) + 12)$ equality constraints, and $nv \times 6 \times (no + 1) + \sum_{i=1}^{nv-1} i \times (no + 1)$ inequality constraints. Figure 2 reports an example of solution for the 3D minimum time UAV formation flying problem with $nv = 25$.

In our analysis, given $N = 3$ benchmark test problems, each of them characterized by a different number of vehicles $nv$, $M = 19$ solvers and algorithms, $K = 50$ randomly generated initial guesses, and $Z = 3$ iterations, a set of $M \times K \times Z$ runs are executed for each test problem. Table 24 reports the number of variables, equality and inequality constraints for the different number of vehicles $nv$. Similarly to the UAV landing problem, some solvers are neglected. In particular, PENLAB, MQA, and SQA are disregarded. For $nv = 10, 25$, SGRA is omitted. The reasons for this choice have been discussed in Section 5.1.

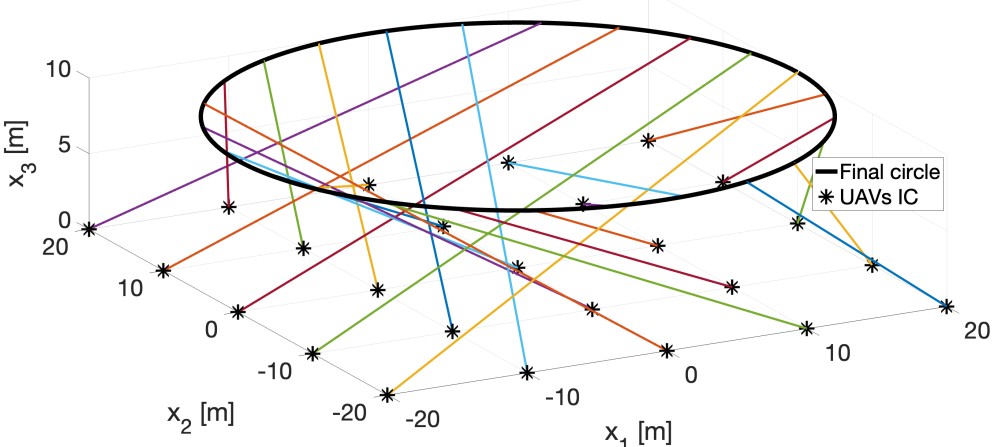

**Figure 2.** Example of solution for UAV Formation Flying problem.

**Table 24.** UAV Formation Flying: problem dimensions and max CPU time threshold.

| $nv$ | 5 | 10 | 25 |
|---|---|---|---|
| Variables | 271 | 541 | 1351 |
| Equality constr. | 240 | 480 | 1200 |
| Inequality const. | 240 | 630 | 2700 |
| $CPU_{max}$ [s] | 20 | 75 | 200 |

Table 25 reports the results for the UAV formation flying problem with $nv = 5$. It can be seen that BARON (auto), BARON (ipopt), BARON (sd), BARON (sqp), FMINCON (sqp), FMINCON (sqp-legacy), KNITRO (active-set), KNITRO (interior-point/D), and KNITRO (sqp) reach 100% convergence rate and similar accuracy in terms of mean error and variance. Almost all the algorithms have similar CPU time, except SNOPT and SOLNP that are one order of magnitude faster and slower, respectively. APSO, MMA/GCMMA, MIDACO, and SGRA instead are not able to find any successful solutions that satisfy the converging threshold conditions in terms of maximum error $E_{max}$. The UAV formation flying problem solutions with $no = 10$ reported in Table 26 show a similar situation, except that now KNITRO (active-set) is no more able to reach full convergence rate, and in general the computational times are increased. FMINCON (interior-point) is not able to converge to any acceptable solutions, together with the same solvers mentioned in the previous case. Regarding the UAV formation flying problem solutions with $nv = 25$ reported in Table 27, all the BARON algorithms have the highest convergence rate and the lowest CPU time. Between all the KNITRO algorithms, instead, only KNITRO (interior-point/D) is able to maintain a quite high convergence rate, probably due to its capability to deal with large-scale problems. It is interesting to note that also KNITRO (interior-point/CG) should be able to deal with large-scale problems, but its performances are not as good as KNITRO (interior-point/D). This can be due to the different way the KKT system is solved by this particular solver [54]. Furthermore, the presence of non-convex collision avoidance constraints in the UAV formation flying problem, together with randomly generated initial conditions, negatively affect the performances of SNOPT, as shown in [76].

**Table 25.** UAV formation flying problem solution for $nv = 5$.

| Ranking | Solver | $\bar{E}$ [%] | $\bar{\sigma}$ [%] | $\bar{\gamma}$ [%] | $\overline{CPU}$ [s] |
|---|---|---|---|---|---|
| 1 | BARON (auto) | $3.681 \times 10^{-4}$ | $7.563 \times 10^{-8}$ | **100.0** | 0.755 |
| 2 | BARON (ipopt) | $3.681 \times 10^{-4}$ | $7.563 \times 10^{-8}$ | **100.0** | 0.795 |
| 3 | BARON (sd) | $3.681 \times 10^{-4}$ | $7.563 \times 10^{-8}$ | **100.0** | 0.727 |
| 4 | BARON (sqp) | $3.681 \times 10^{-4}$ | $7.560 \times 10^{-8}$ | **100.0** | 0.734 |
| 5 | FMINCON (sqp) | $3.681 \times 10^{-4}$ | $7.563 \times 10^{-8}$ | **100.0** | 0.397 |
| 6 | FMINCON (sqp-legacy) | $3.681 \times 10^{-4}$ | $7.563 \times 10^{-8}$ | **100.0** | 0.446 |
| 7 | KNITRO (active-set) | $3.681 \times 10^{-4}$ | $7.563 \times 10^{-8}$ | **100.0** | 0.761 |
| 8 | KNITRO (interior-point/D) | $3.681 \times 10^{-4}$ | $7.564 \times 10^{-8}$ | **100.0** | 0.498 |
| 9 | KNITRO (sqp) | $3.681 \times 10^{-4}$ | $7.563 \times 10^{-8}$ | **100.0** | 0.680 |
| 10 | KNITRO (interior-point/CG) | $3.678 \times 10^{-4}$ | $7.963 \times 10^{-8}$ | **98.0** | 0.862 |
| 11 | SNOPT | $3.632 \times 10^{-4}$ | $7.401 \times 10^{-8}$ | **96.0** | 0.071 |
| 12 | FMINCON (interior-point) | 1.400 | 2.390 | **76.0** | 0.594 |
| 13 | FMINCON (active-set) | $2.811 \times 10^{-4}$ | $3.926 \times 10^{-8}$ | **30.0** | 0.535 |
| 14 | SOLNP | $1.125 \times 10^{-3}$ | $8.907 \times 10^{-7}$ | **34.0** | 2.846 |
| - | APSO | $>E_{max}$ | - | **-** | - |
| - | GCMMA | $>E_{max}$ | - | **-** | - |
| - | MIDACO | $>E_{max}$ | - | **-** | - |
| - | MMA | $>E_{max}$ | - | **-** | - |
| - | SGRA | $>E_{max}$ | - | **-** | - |

**Table 26.** UAV formation flying problem solution for $nv = 10$.

| Ranking | Solver | $\bar{E}$ [%] | $\bar{\sigma}$ [%] | $\bar{\gamma}$ [%] | $\overline{CPU}$ [s] |
|---|---|---|---|---|---|
| 1 | BARON (auto) | $3.492 \times 10^{-4}$ | $6.299 \times 10^{-8}$ | **100.0** | 1.481 |
| 2 | BARON (ipopt) | $3.492 \times 10^{-4}$ | $6.299 \times 10^{-8}$ | **100.0** | 1.530 |
| 3 | BARON (sd) | $3.492 \times 10^{-4}$ | $6.299 \times 10^{-8}$ | **100.0** | 1.415 |
| 4 | BARON (sqp) | $3.492 \times 10^{-4}$ | $6.299 \times 10^{-8}$ | **100.0** | 1.432 |
| 5 | FMINCON (sqp) | $3.492 \times 10^{-4}$ | $6.299 \times 10^{-8}$ | **100.0** | 1.785 |
| 6 | FMINCON (sqp-legacy) | $3.492 \times 10^{-4}$ | $6.299 \times 10^{-8}$ | **100.0** | 2.047 |
| 7 | KNITRO (interior-point/D) | $3.492 \times 10^{-4}$ | $6.299 \times 10^{-8}$ | **100.0** | 3.172 |
| 8 | KNITRO (sqp) | $3.492 \times 10^{-4}$ | $6.299 \times 10^{-8}$ | **100.0** | 9.651 |
| 9 | KNITRO (interior-point/CG) | $3.400 \times 10^{-4}$ | $6.397 \times 10^{-8}$ | **96.0** | 6.333 |
| 10 | KNITRO (active-set) | $3.233 \times 10^{-4}$ | $4.785 \times 10^{-8}$ | **96.0** | 10.044 |
| 11 | SNOPT | $8.069 \times 10^{-4}$ | $1.035 \times 10^{-5}$ | **96.0** | 2.811 |
| 12 | FMINCON (active-set) | $3.803 \times 10^{-4}$ | $4.417 \times 10^{-8}$ | **36.0** | 1.963 |
| 13 | SOLNP | $1.064 \times 10^{-3}$ | $8.218 \times 10^{-7}$ | **12.0** | 16.013 |
| - | APSO | $>E_{max}$ | - | **-** | - |
| - | FMINCON (interior-point) | $>E_{max}$ | - | **-** | - |
| - | GCMMA | $>E_{max}$ | - | **-** | - |
| - | MIDACO | $>E_{max}$ | - | **-** | - |
| - | MMA | $>E_{max}$ | - | **-** | - |

**Table 27.** UAV formation flying problem solution for $nv = 25$.

| Ranking | Solver | $\bar{E}$ [%] | $\bar{\sigma}$ [%] | $\bar{\gamma}$ [%] | $\overline{CPU}$ [s] |
|---|---|---|---|---|---|
| 1 | BARON (auto) | $1.953 \times 10^{-4}$ | $6.562 \times 10^{-8}$ | **96.0** | 5.782 |
| 2 | BARON (ipopt) | $1.953 \times 10^{-4}$ | $6.562 \times 10^{-8}$ | **96.0** | 13.923 |
| 3 | BARON (sd) | $1.953 \times 10^{-4}$ | $6.563 \times 10^{-8}$ | **96.0** | 5.052 |
| 4 | BARON (sqp) | $1.953 \times 10^{-4}$ | $6.562 \times 10^{-8}$ | **96.0** | 5.299 |
| 5 | KNITRO (interior-point/D) | $3.117 \times 10^{-3}$ | $3.899 \times 10^{-4}$ | **90.0** | 119.081 |
| 6 | FMINCON (sqp) | $2.506 \times 10^{-2}$ | $2.302 \times 10^{-2}$ | **74.0** | 48.901 |
| 7 | FMINCON (sqp-legacy) | $2.506 \times 10^{-2}$ | $2.302 \times 10^{-2}$ | **74.0** | 54.861 |
| 8 | SNOPT | $2.083 \times 10^{-4}$ | $7.497 \times 10^{-8}$ | **44.0** | 45.408 |
| 9 | KNITRO (sqp) | $2.025 \times 10^{-4}$ | $7.837 \times 10^{-8}$ | **22.0** | 133.070 |
| 10 | FMINCON (active-set) | $4.678 \times 10^{-5}$ | $2.198 \times 10^{-9}$ | **10.0** | 21.308 |
| 11 | KNITRO (active-set) | $3.933 \times 10^{-5}$ | $4.017 \times 10^{-9}$ | **6.0** | 65.720 |
| 12 | KNITRO (interior-point/CG) | 1.484 | 0.000 | **2.0** | 165.145 |
| - | APSO | $>E_{max}$ | - | **-** | - |
| - | FMINCON (interior-point) | $>E_{max}$ | - | **-** | - |
| - | GCMMA | $>E_{max}$ | - | **-** | - |
| - | MIDACO | $>E_{max}$ | - | **-** | - |
| - | MMA | $>E_{max}$ | - | **-** | - |
| - | SOLNP | $>E_{max}$ | - | **-** | - |

## 6. Conclusions

In this paper, we provide an explicit comparison of a set of NLP solvers for the solution of unconstrained and constrained NLP problems. Because of its widespread use among research groups, both in academia and the private sector, we have used MATLAB as a common implementation platform. The benchmark analysis aims to compare popular solvers which are readily available in MATLAB, a few gradient descent methods that have been extensively used in the literature, and a particle swarm optimization in terms of accuracy, convergence rate, and computational time. In addition, three different implementation scenarios per each solver are taken into consideration, namely, plug and play (P&P), high accuracy (HA), and quick solution (QS). With this is mind, at first, each solver has been tested on a selection of constrained and unconstrained standard benchmark problems with up to thirty variables and a up to nine scalar constraints. Results for the unconstrained problems show that BARON is the algorithm that delivers the best convergence rate and accuracy but it is the slowest. PENLAB is the algorithm that has the best trade-off between accuracy, convergence rate, and speed. For the constrained NLP problems, again, BARON showcases exceptional accuracy and convergence rate, yet it falls within the slower range of algorithms. FMINCON, KNITRO, and SNOPT are the ones that are able to deliver a fair compromise of accuracy, convergence rate, and speed. Then, we have tested each solver to solve two large scale real-world minimum-time UAV optimal path planning problems for UAV landing and formation flying. Results for the UAV landing problems show that BARON and SNOPT deliver the best convergence rate and accuracy, with SNOPT being the fastest solver. Some solvers instead are not able to converge to any acceptable solutions. Finally, results for UAV formation flying problem show that overall BARON still reaches the best convergence rate and accuracy, but also the lowest CPU time. KNITRO and FMINCON follow closely behind BARON.

**Author Contributions:** Conceptualization, M.C. and G.L.; methodology, G.L., V.C. and M.C.; software, G.L. and K.G.; validation, G.L. and K.G.; formal analysis, G.L., V.C. and M.C.; investigation, M.C. and G.L.; resources, G.L., K.G., V.C. and M.C.; data curation, G.L.; writing—original draft preparation, M.C. and G.L.; writing—review and editing, G.L., K.G., V.C. and M.C.; visualization, G.L.; supervision, V.C. and M.C.; project administration, M.C. and G.L.; funding acquisition, M.C. and V.C. All authors have read and agreed to the published version of the manuscript.

**Funding:** This research was supported by Amazon (Amazon Research Award 2021), by the Office of Naval Research (grants N000142212634 and N000142112091), and by the National Science Foundation (grant 2136298).

**Data Availability Statement:** Data available on request from the authors.

**Conflicts of Interest:** The authors declare no conflict of interest.

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
