# Peer review of "Comparative Analysis of Nonlinear Programming Solvers: Performance Evaluation, Benchmarking, and Multi-UAV Optimal Path Planning"

_drones, doi:10.3390/drones7080487_

Round 1

Reviewer 1 Report

This paper gives a set of guidelines to select a solver for the solution of nonlinear programming problems. the capability of each solver to tackle two large scale minimum-time UAV optimal path planning problems is evaluated. This research is necessary and meaningful. There still exist several minor issues that need to be addressed. The details are as follows.

The specific path planning problem of UAV should be described in detail, including applicable scenarios, constraints, and limitations.

Reviewer 2 Report

Authors made a notorious work and research, with a complete comparison of algorithms regarding nonlinear programming solvers. Although, my concerns go to the particular theme of multi-UAV optimal path planning, which lacks definition and discussion along the work. Below are my comments to the authors, which should be considered to improve the overall quality of the paper.

1. The abstract should have a more clear view and introduction to the problem background, specially regarding the path planning problem which lacks definition.

2. Besides conclusions have a discussion about the algorithms, I believe that they would be more interesting to the readers if they had a discussion on the significance of the results regarding the problem of path planning, not only on the analytical side.

English is fine.

Reviewer 3 Report

This paper provides a comprehensive comparison of various Nonlinear Programming (NLP) solvers.  The analysis involved standard benchmark problems and two large-scale, minimum-time Unmanned Aerial Vehicle (UAV) optimal path planning problems. The paper adopts the Bernstein polynomial approximations for 3D UAV trajectory planning, turning infinite-dimensional optimization problems into NLP ones. 

- Although the paper is well-written and the contents could be practically interesting, as many solvers are indeed widely applied to solve NLP in both academia and industry, it is still questionable if it fits this journal. 

- There are many large-scale NLP problems in other scenarios. It is suggested to elobrate why UAV-related problems are considered as benchmarks, i.e., are they typical as NLPs? or what makes them special?

- It would be more interesting to investigate the different results between UAV-related problems and other constrainted test problems. For example, why SNOPT performs worse in formation flying problems than it in test problems? Are the gap siginificant? 

-In additional to average the metric, peak/outage performance metrics could also provides some insights for choosing the solvers.

Round 2

Reviewer 2 Report

Authors have answered all my suggestions.

Reviewer 3 Report

The reviewer does not have futher comments.